# Severe Myocardial Dysfunction after Non-Ischemic Cardiac Arrest: Effectiveness of Percutaneous Assist Devices

**DOI:** 10.3390/jcm10163623

**Published:** 2021-08-17

**Authors:** Stéphane Manzo-Silberman, Christoph Nix, Andreas Goetzenich, Pierre Demondion, Chantal Kang, Michel Bonneau, Alain Cohen-Solal, Pascal Leprince, Guillaume Lebreton

**Affiliations:** 1Department of Cardiology, Lariboisière Hospital, Assistance Publique-Hôpitaux de Paris (AP-HP), Denis Diderot University, INSERM UMRS 942, 75010 Paris, France; alain.cohen-solal@aphp.fr; 2Abiomed Europe GmbH, Neuenhofer Weg 3, D-52074 Aachen, Germany; cnix@abiomed.com (C.N.); agoetzenich@abiomed.com (A.G.); 3Department of Cardiovascular and Thoracic Surgery, Institute of Cardiology, Pitié-Salpêtrière Hospital, Assistance Publique-Hôpitaux de Paris (AP-HP), Sorbonne University, 47-83 Boulevard de l’Hôpital, 75013 Paris, France; pierre.demondion@aphp.fr (P.D.); pascal.leprince@aphp.fr (P.L.); guillaume.lebreton@aphp.fr (G.L.); 4XP-MED, 78100 Saint Germain en Laye, France; chantal.kang@free.fr (C.K.); michel.bonneau24@orange.fr (M.B.)

**Keywords:** acute heart failure, cardiogenic shock, physiopathology, mechanical circulatory support, organ replacement therapies, extracorporeal membrane oxygenation, left ventricular assist device

## Abstract

Introduction: Despite the improvements in standardized cardiopulmonary resuscitation, survival remains low, mainly due to initial myocardial dysfunction and hemodynamic instability. Our goal was to compare the efficacy of two left ventricular assist devices on resuscitation and hemodynamic supply in a porcine model of ventricular fibrillation (VF) cardiac arrest. Methods: Seventeen anaesthetized pigs had 12 min of untreated VF followed by 6 min of chest compression and boluses of epinephrine. Next, a first defibrillation was attempted and pigs were randomized to any of the three groups: control (*n* = 5), implantation of an percutaneous left ventricular assist device (Impella, *n* = 5) or extracorporeal membrane oxygenation (ECMO, *n* = 7). Hemodynamic and myocardial functions were evaluated invasively at baseline, at return of spontaneous circulation (ROSC), after 10–30–60–120–240 min post-resuscitation. The primary endpoint was the rate of ROSC. Results: Only one of 5 pigs in the control group, 5 of 5 pigs in the Impella group, and 5 of 7 pigs in the ECMO group had ROSC (*p* < 0.05). Left ventricular ejection fraction at 240 min post-resuscitation was 37.5 ± 6.2% in the ECMO group vs. 23 ± 3% in the Impella group (*p* = 0.06). No significant difference in hemodynamic parameters was observed between the two ventricular assist devices. Conclusion: Early mechanical circulatory support appeared to improve resuscitation rates in a shockable rhythm model of cardiac arrest. This approach appears promising and should be further evaluated.

## 1. Introduction

Cardiac arrest is a major cause of mortality and morbidity in Western countries. Each year, more than 350,000 people in the United States die suddenly [1]. Coronary artery disease (CAD) is a major cause of out-of-hospital cardiac arrest (OHCA) [2,3]. However, in 30% of patients, cardiac arrest is neither associated with CAD nor with an obvious extracardiac cause [4]. Despite recent advances in public education and resuscitation processes, only a few patients are transported alive to the hospital, and even less are discharged from the hospital without sequelae. Even among patients arriving at the hospital, survival rates remain low, ranging from 21–33%, and have not improved in recent years [5,6,7]. Successful treatment of cardiac arrest includes the rapid initiation of basic life support measures, including cardiopulmonary resuscitation (CPR). Immediate institution of such efforts can result in resuscitation rates as high as 30–40% [8,9].

In the cardiovascular post-resuscitation syndrome, cardiac filling pressures temporarily increase and the cardiac index dramatically decline [7,10]. Even after effective resuscitation and recovery of myocardial blood flow, ventricular functions are markedly depressed [10,11,12]. These findings underscore the need to restore spontaneous circulation via rapid and aggressive supportive therapy, with the expectation that part of myocardial stunning will recover. The need of an early-onset invasive strategy for OHCA was the central premise of the study evaluating the feasibility and safety of pre-hospital insertion of extra-corporeal membrane oxygenation (ECMO) [13]. Recently, the ARREST study reported improved survival in ECMO-facilitated resuscitation [14], as well as a Danish multicenter study [15].

In patients with acute heart failure, percutaneous mechanical circulatory support (MCS) devices are increasingly used [16]. Durable ventricular assist devices are complex, require surgical implantation, and are expensive. Thus, their use in the setting of cardiac arrest is not indicated given the uncertain prognosis and urgent nature of care. Percutaneous MCS devices are a viable option in cardiac arrest, owing to the rapid initiation of support and ease of use. The feasibility and superiority of ECMO compared to conventional CPR in adults with in-hospital cardiac arrest have been demonstrated [17,18]. Nevertheless, the use of ECMO involves large cannula, complex vascular access, and is associated with ischemic and infectious complications [19].

The Impella CP percutaneous assist device (Abiomed Inc., Danvers, MA, USA) is a valuable alternative to ECMO.

We previously reported that left-sided percutaneous support with Impella 2.5 was feasible in post-resuscitation shock with a high percentage of good neurological recovery [20]. These results were confirmed in a retrospective multicenter analysis [21] and a Danish multicenter study recently confirmed higher survival rate in cardiac arrested patient in adherence to national consensus selection criteria [15].

As ECMO has already been investigated as a rescue-device in refractory cardiac arrest with early implantation of the device in the field [13,22,23], the objective of this study was to compare the two percutaneous mechanical circulatory support devices that can be deployed early: The Impella CP versus ECMO. We compared ROSC obtention, hemodynamic parameters, myocardial recovery, and peripheral organ damage in a porcine model of ventricular fibrillation (VF) cardiac arrest, as the best model in cardiac arrest study [12,24,25].

## 2. Material and Methods

### 2.1. Animal Preparation

All procedures were conducted in accordance with the basic principles for the care and use of animals based on the Helsinki Declaration and were approved by the appropriate governmental institution. Seventeen male domestic pigs at 4 months of age (weighing 34.6 ± 4.6 kg) were studied in a model of untreated VF followed by mechanical support alone to mimic out-of-hospital cardiac arrest treatment.

Under anesthesia (Isoflurane) and mechanical endotracheal ventilation, a Swan-Ganz catheter was placed through the left external jugular vein and a 7F sheath was inserted in the right external jugular vein to place the electrodes for the required pacing [12]. A 6F sheath was introduced in the right carotid artery to place a pig tail catheter into the left ventricular cavity for the measurement of LV pressures and the angiography for LVEF estimation [12,26].

The left femoral artery was used as an access for continuous invasive pressure measurement.

The Impella was placed via the right femoral artery through a 13F sheath.

In the ECMO group, the arterial 15F cannula was placed in the right femoral artery and the venous cannula (21 F) was inserted through the left femoral vein.

ECG, pulse oximetry, and rectal temperature were continuously monitored using a standard patient monitor.

Peripheral tissue oxygen saturation (StO2) levels were measured using near infrared spectroscopy (NIRS) with an INVOS^®^ Cerebral/Somatic Oximeter (Somanetics Corporation, Troy, MI, USA). The sensor was attached to the forehead.

Unfractioned heparin (5000 UI IV) was given as a bolus after sheath placement followed by additional boluses of 2500 UI IV to maintain an activated clotting time between 180 and 250 s.

Initial rapid IV infusion of 1000 mL of normal saline was given intravenously, followed by a continuous IV drip of 200 to 500 mL/h to reach and maintain a central venous pressure of 3–7 mmHg. Additional saline as rapid IV boluses of 100–250 mL was administered during the period of assistance.

Prior to the experimental protocol, a stabilization period of one hour was used to let the animal recover from instrumentation. Simple randomization was performed to define group allocation for each animal as experimentations were planned.

### 2.2. Experimental Protocol

VF was electrically induced by pacing the right ventricle at rates of 400–600 beats/min as confirmed by electrocardiography (ECG) and loss of the aortic pressure pulsations. Anesthesia and ventilation were discontinued during the arrest period. Cardiopulmonary resuscitation efforts were initiated after 12 min of untreated VF. The 12-min untreated VF period was chosen so that the myocardial insult would be severe enough to allow discrimination of outcomes among the three cardiac assistance groups.

Resuscitation [11,27,28,29] efforts included standard Basic Life Support, such as external chest compressions (by a pneumatically driven automatic piston device: Pneumatic Compression Controller, Ambu International, Glostrup, Denmark) and positive pressure ventilation with room air (FiO_2_ of 21%). The piston device provided a 4–6 cm compression, adjusted on the pig anatomy, at a rate of 100/min. The tidal volume was maintained at 10 mL/Kg. Epinephrine was administered as a 1-mg intravenous bolus at the initiation of CPR and 3 min after initiation. After 6 min of CPR, one external defibrillation (Zoll) was attempted with a 200 J biphasic shock, and additional defibrillation shocks, still biphasic 200 J, were delivered every 3 min preceded by an intravenous bolus of 1 mg epinephrine, if needed. After 6 min of CPR and the first external electric shock, cardiac assistance was started according to the group assignment (Figure 1). For logistic reasons, assignment to the groups was decided at the beginning of the experiment.

Resuscitation efforts were continued until ROSC was achieved or for a total of 15 min after delivery of the first external electric shock.

After ROSC was achieved, animals were mechanically ventilated, with supplemental oxygen if needed to obtain an arterial saturation greater than 90%. During the immediate post defibrillation stage, amiodarone, epinephrine or atropine was used, according to the advanced cardiac life support guidelines [7,28,29,30,31,32,33,34,35].

All groups received therapeutic hypothermia immediately after ROSC by infusion of 1.5 L of chilled saline with a target temperature between 34 and 36 °C. When the animals were stabilized, they were observed under general anesthesia and isoflurane was titrated to maintain adequate anesthesia and analgesia during the whole period of evaluation. 

Animals remained instrumented for measurement of hemodynamic parameters. Fluid loading by saline was infused if required, likewise epinephrine boluses and, if needed infusion, depending on the hemodynamic parameters.

After the final data collection was completed, animals were euthanized by an intravenous solution of Doléthal^®^ (Vétoquinol S.A., Lure, France) and necropsy studies were performed by a veterinarian. All the animals involved in the study were included in the analysis, no exclusion criteria were applied.

### 2.3. Data Analysis

In vivo evaluations were performed at baseline, before cardiac arrest, after successful resuscitation, and before initiation of hemodynamic support (if available), then 10–30–60–120–240 min after hemodynamic support.

The primary outcome of this study was the rate of ROSC.

Secondary outcomes were:(1)LV function (LVEF) after 4 h post-resuscitation measured by fluoroscopy (RAO 30° incidence) using an injection of 12 mL contrast medium (Visipaque^®^, GE Healthcare SAS, Boston, MA, USA) [26](2)Myocardial function: Left ventricular end-diastolic pressure was measured by the pigtail, cardiac outflow, systolic ejection volume (mL/min), left ventricular systolic and diastolic volumes measured by fluoroscopy (RAO 30° incidence), and calculated using the standard software of the angiography suite;(3)Hemodynamic efficacy assessed based on systolic, mean and diastolic arterial pressures, central venous pressure, and pulmonary arterial pressure.

### 2.4. Statistical Analysis

Values are expressed as mean ± SD or number (%), as appropriate. Due to the small sample size and the likelihood of high heterogeneity, the experiments were planned as a hypothesis-generating study. The primary endpoint of successful ROSC was statistically evaluated using a Fisher’s exact test for a 2 × 3 contingency table. All secondary endpoints are merely descriptive. Except for the estimation of rate of ROSC, the control group was not included in the comparative analysis due to its poor outcome. In fact, within the control group, only one animal out of five was successfully resuscitated.

Non-normal distribution of the data was confirmed using the Shapiro test. Differences between groups were assessed using exact Wilcoxon-Mann-Whitney Test. All statistical analyses were performed using the R-statistical software (R Foundation for Statistical Computing, Vienna, Austria). A 2-sided *p*-value < 0.05 was considered statistically significant.

## 3. Results

### 3.1. Resuscitation, ROSC and Survival

The characteristics of the resuscitation between the 3 groups are presented in Table 1. As shown in Figure 2A, pigs receiving hemodynamic support showed a higher rate of ROSC, with a 100% (5/5) success rate in the Impella group, 71.4% (5/7) in the ECMO group, and 20% (1/5) in the control group (*p* = 0.044). In pigs achieving ROSC, the duration to ROSC was shorter in the ECMO group. Nevertheless, two pigs treated with ECMO died during this period and post-mortem autopsy revealed hemorrhagic complications related to arterial (carotid or iliac) injuries. A third pig in the ECMO group also died despite initial ROSC within a few minutes due to iliac arterial bleeding. All pigs in the Impella group survived the initial phase of the experiment but 2 pigs died between 120–240 min. In one of the pigs, asystole occurred two hours after ROSC, CPR failed, and the post-mortem revealed a hemomediastinum, possibly a consequence of the mechanical chest compression. In the second pig, VF occurred after four hours of Impella support and multiple electrical cardioversion attempts failed. The post-mortem examination revealed severe cardiac hypertrophy as a pre-existing pathology and the presumable cause for the re-occurring and resistant arrhythmia.

As can be expected, the amount of epinephrine administered was lower in the mechanically assisted pigs. Despite a higher initial survival rate within the Impella group, no difference in survival was observed between the two assist devices at the end of the experiment (Figure 2B).

### 3.2. Mechanical Circulatory Support and Hemodynamic

The Impella system maintained sufficient mean aortic pressure (Figure 3A) during the entire experiment despite reduced flow rate than ECMO (Appendix A). Although the total flow was measured using a Swan-Ganz catheter (thermodilution method), these measurements are typically highly unreliable in the setting of mechanical circulatory support.

Even though pigs in the ECMO group showed a sustained reduction in LVEF in the early phase post-ROSC, there was no significant difference in LVEF between the two cardiac assist devices at 4 h post-resuscitation (Figure 3B).

Table 2 shows the different measurements of the myocardial function in the two groups receiving hemodynamic support.

No significant difference in hemodynamic parameters was observed between the support devices (Table 3). There was no difference in the hemodynamic variables between the devices at baseline. At the time of ROSC, the ECMO showed a trend towards better hemodynamics with higher MAP 131 vs. 89 mmHg (*p* = 0.15). Pulmonary arterial pressure was not different between the devices throughout the experiment.

As per the protocol, no correction of the acid-base balance was initiated during the 4 h of mechanical support. Blood gas analyses (Table 3) showed a numerically increased paO_2_ at the oxygenation membrane of the ECMO. The lactate levels did not differ between the two groups.

## 4. Discussion

This animal study demonstrated that the prompt initiation of percutaneous cardiac assistance during experimental cardiac arrest increased the rate of ROSC with a 100% success with the Impella device in the setting of prolonged VF. Compared to ECMO, no difference in hemodynamic parameters was observed with Impella.

### 4.1. Effectiveness of Prompt Cardiac Assistance

Cardiac assist devices are frequently used as a rescue therapy following failure of standard CPR. Data from clinical studies evaluating use of ECMO in the setting of OHCA are limited and they report low survival rates and disappointing neurologic outcomes [36,37]. However, ECMO is still useful in situations requiring high flow support and optimal oxygenation [15,38]. Schmidt et al. suggested 10 situations in which use of ECMO is often futile and included a case series of critically ill patients with long low-flow periods, lasting more than 30 min and more [39]. Recently, Lamhaut et al. demonstrated the feasibility and safety of initiation of pre-hospital ECMO for refractory CA in the field by an experienced team [13]. Their propensity analysis [22] also suggested that prehospital ECMO in a selected population may improve survival and neurological outcome, in particular if initiated earlier.

Our animal study supports this new paradigm of prompt initiation of cardiac support. The delay of 12 min was chosen to cause a significant myocardial dysfunction and could also reflects the clinical setting with a medium delay of emergency team arrival. The delay to start the assistance device would add approximately another ten minutes by experienced teams.

The use of Impella would involve a less invasive approach with lower vascular complication rates than ECMO, yet would require placement using radiography, thus currently limiting its use in the field.

In a similar animal experimental setup, Derwall et al. showed increased survival and improved outcome by CPR using Impella instead of chest compression [24]. Standard CPR with chest compression alone, even of high quality by a mechanical thumper demonstrated disappointing results [40]. The current growing evidence of epinephrine´s ambivalent results, showing higher rates of sustained ROSC but without any improvement in favorable neurological outcomes [41,42] supports the need for new approaches in resuscitation, in particular, a shift toward higher use of mechanical assist devices. The Danish multicenter study reported improved survival rate and good neurological outcome in a population of patient treated by mechanical circulatory support for OHCA [15]

### 4.2. Hemodynamic Support

Left ventricular assistance with Impella has been shown to have beneficial effects on coronary perfusion and the effective unloading of the LV reduced myocardial injury after reperfusion [43,44].

Based on these findings, active unloading with Impella in the acute phase may improve myocardial recovery after the initial stunning compared to the increase in afterload exerted by ECMO. Kern et al. showed complete recovery of myocardial stunning by 48 h after sustained VF [12]. However, in the animal model used in this study, measurements such as LVEF were taken only until 4 h after ROSC; nevertheless, the LVEF was also already improving by 4 h after ROSC.

Early after ROSC, LVEF in the Impella group seemed to be preserved during the first 2 h but started to decrease later. Within our experimental setting, the favorable effect of unloading by the Impella pump was counterbalanced by suction events that necessitated a reduction of flow. Automatic adaptation of flow, as available in the console for clinical use, was not effective in our setting. On the other hand, ECMO showed a reduction in LVEF during the initial phase, presumably due to an increase in afterload, and thus a delayed recovery. The young and healthy hearts might be able to compensate for the increase in the afterload over time with recovery within one hour post-ROSC. However, this increase in afterload might have severe consequences in the setting of heart condition with pre-existing comorbidities.

The expected reduction of LVEDP following LV unloading was not confirmed in our experiment. In fact, the pulmonary pressures did not decrease with unloading as shown previously in other experiments [24].

### 4.3. Limitations

We recognize several limitations in this hypothesis generating study without any sample size calculation. First, the pigs in the control group representing the clinical situation of cardiac arrest treatment without mechanical support had a very low rate of ROSC, thus limiting the comparisons. Nevertheless, the low rate of immediately successful resuscitation was comparable to the one observed in the literature [24]. The poor survival in the control group convinced us to not expose additional pigs to this condition.

Second, due to short-term evaluations, we did not perform clinical and neurological assessments. However, our measurement until four hours of support showed trends that can be interpreted. Invasive measurements of baseline hemodynamic parameters could have compromised survival and ROSC success rates.

Third, in order to mimic the constraints in the setting of OHCA, all biological measurements were analyzed after the experiment. As a result, ventilation was not adapted according to the blood gases for any of the groups, resulting in differences between the groups and hyperventilation in the ECMO group. Likewise, initial ventilation with room air (FiO_2_ of 21%) was chosen to be closer to the initial ventilation conditions “on the field”. A more intensive oxygenation would have been more efficient. We acknowledged that the resuscitation did not completely reflect current guideline recommendations.

Fourth, the complications observed in the ECMO group deserve consideration. The highly invasive procedure of cannulation in an emergency situation increases the risk of arterial misplacement and dissection, even in the relatively controlled setting of an animal experiment. This procedural aspect may partially explain the worse results observed in OHCA patients [13,36] compared to in-hospital cardiac arrest patients assisted with an ECMO [17]. The placement of the Impella requires vascular access and in patients with cardiac arrest, puncture of the femoral artery remains feasible and can be performed using ultrasound guidance [45]. 

Finally, all the pigs had normal myocardium without coronary artery disease and one can speculate that the myocardial stunning after prolonged VF could be even more marked in aged hearts with concomitant atherosclerotic disease and prompt LV support in this setting could improve outcomes significantly.

## 5. Conclusions

In this porcine model of VF cardiac arrest, early assistance with cardiac devices was associated with improved rates of return of spontaneous circulation. No significant difference was observed between Impella and ECMO percutaneous circulatory support, may be due to small sample size. The use of percutaneous circulatory support in the setting of cardiac arrest to increase the rate of ROSC is a promising strategy. Further clinical randomized studies are warranted to delineate the safety and effectiveness of these devices specifically.

## Figures and Tables

**Figure 1 jcm-10-03623-f001:**
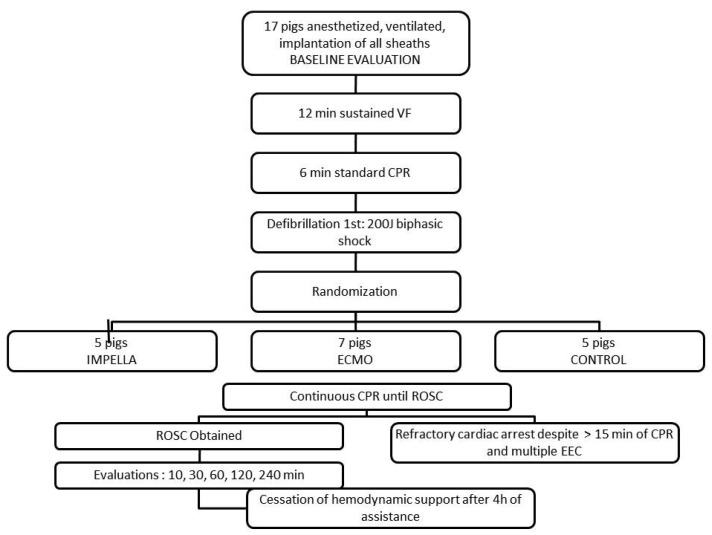
Study design.

**Figure 2 jcm-10-03623-f002:**
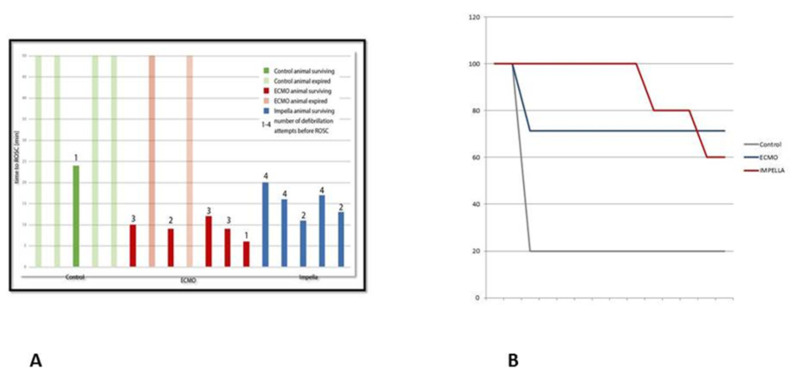
ROSC delay and achievement (**A**) and Survival (**B**).

**Figure 3 jcm-10-03623-f003:**
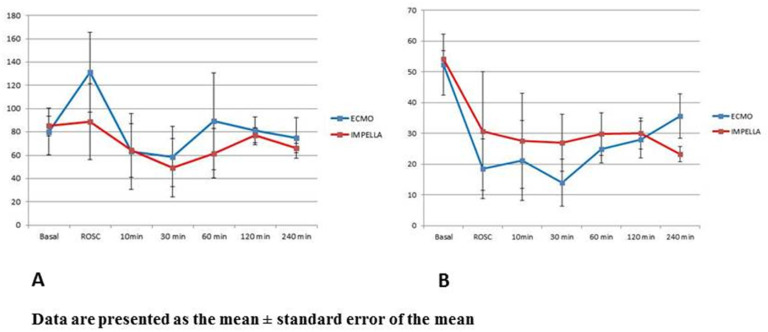
Mean aortic pressure (**A**) and left ventricle ejection fraction (**B**).

**Table 1 jcm-10-03623-t001:** Comparison of resuscitation parameters.

	CONTROL *n* = 5	ECMO *n* = 7	IMPELLA *n* = 5	*p* Value ECMO vs. IMPELLA
Epinephrine total dose mg	6.5 ± 1.9	3.6 ± 0.8	3.2 ± 0.8	0.623
External shock (*n*)	4 ± 2.4	2.4 ± 0.8	3.2 ± 1.1	0.2
Total VF Duration (min)	19 ± 1.7	20.7 ± 2.9	25 ± 4.8	0.13
Total duration chest compressions	25 ± 11.4	10.4 ± 5.8	13.4 ± 5.4	0.28
ROSC before initiation of assistance *n* (%)	1 (20)	1 (14.3)	1 (20)	1
Success *n* (%)	1(20)	5 (71.4)	5 (100)	0.6

Mean ± Standard deviation.

**Table 2 jcm-10-03623-t002:** Myocardial function data.

	Basal ECMO = 7 Impella = 5	ROSC ECMO = 5 Impella = 5	10 min ECMO = 5 Impella = 5	30 min ECMO = 4 Impella = 5	60 min ECMO = 4 Impella = 5	120 min ECMO = 4 Impella = 4	240 minECMO = 4 Impella = 3
LVEDV							
ECMO =	45.2 ± 11.2	45.9 ± 13.5	49.6 ± 12.1	56.3 ± 2.9	54.8 ± 8.9	49.5 ± 6.4	41.9 ± 15.8
Impella =	53.8 ± 18.6	35.4 ± 12.8	31.9 ± 17.7	42.7 ± 15.5	44.7 ± 13.1	51.9 ± 15.8	62.6 ± 13.5
LVESV							
ECMO =	21.0 ± 7.8	38.6 ± 15.0	40.2 ± 15.9	48.6 ± 6.5	41.1 ± 8.8	36.0 ± 7.2	26.9 ± 7.2
Impella =	24.6 ± 8.7	25.7 ± 15.9	28.48 ± 9.5	31.6 ± 14.0	31.7 ± 11.3	40.3 ± 15.4	48.1 ± 15.4
LVEDP (mmHg)							
ECMO =	5.86 ± 7.9	11.6 ± 16.5	13.0 ± 10.1	14.5 ± 5.3	3.75 ± 1.5	4.5 ± 4.0	4.5 ± 5.8
Impella =	8.6 ± 6.3	20.2 ± 17.9	16.4 ± 12.4	13.25 ± 9.4	11.6 ± 7.8	8.75 ± 7.7	12.67 ± 10.8
Cardiac Output (L/min)							
ECMO =	6.20 ± 2.36	6.15 ± 3.53	5.78 ± 3.36	3.70 ^†^	4.0 ± 2.12	5.56 ± 0.83	7.73 ± 2.75
Impella =	5.98 ± 1.28	4.03 ± 3.21	4.47 ± 1.42	2.10 ^†^	4.3 ± 2.49	7.68 ± 2.09	5.73 ± 1.09

Mean ± Standard deviation; ^†^: missing data, SD not available.

**Table 3 jcm-10-03623-t003:** Comparison of hemodynamics and blood gas data.

	Basal ECMO = 7 Impella = 5	ROSC ECMO = 5 Impella = 5	10 min ECMO = 5 Impella = 5	30 min ECMO = 4 Impella = 5	60 min ECMO = 4 Impella =5	120 min ECMO = 4 Impella = 4	240 min ECMO = 4 Impella = 3
HR (bpm)							
ECMO	141 ± 35	131 ± 26	128 ± 16	127 ± 16	147 ± 21	150 ± 12	147 ± 35
Impella	118 ± 21	131 ± 15	157 ± 29	120 ± 34	126 ± 43	155 ± 18	163 ± 6
MAP (mmHg)							
ECMO =	84.14 ± 18.2	137.8 ± 33.8	68.75 ± 34.8	67.33 ± 23.8	96.25 ± 44.5	83.25 ± 12.5	79.75 ± 15.4
Impella =	85.20 ± 8.6	88.8 ± 32.6	64.20 ± 23.1	49.25 ± 24.8	61.6 ± 21.2	77.25 ± 6.2	66.33 ± 4.0
SPAP (mmHg)							
ECMO =	29.9 ± 5.9	27.8 ± 14.8	24.4 ± 20.7	21.0 ± 11.7	42.0 ± 26.2	35.0 ± 14.8	40.5 ± 17.6
Impella =	25.8 ± 1.8	40.6 ± 12.6	29.0 ± 7.8	35.8 ± 12.0	36.6 ± 10.7	42.8 ± 5.7	36.7 ± 9.3
CVP (mmHg)							
ECMO =	6.0 ± 4.3	3.6 ± 3.4	3.2 ± 4.3	4.0 ± 2.9	3.75 ± 2.2	4.0 ± 4.2	5.0 ± 3.4
Impella =	5.8 ± 3.6	12.2 ± 4.4	10.6 ± 4.9	8.75 ± 2.9	7.4 ± 3.6	5.5 ± 2.6	4.3 ± 2.1
PaO_2_ (mmHg)							
ECMO =	485.6 ± 83.4	558.8 ± 28.3	435.6 ± 215.7	349.7 ± 164.2	501.7 ± 104.4	456.9 ± 69.9	373.0 ± 86.9
Impella =	437.9 ± 61.4	238.7 ± 138.2	227.9 ± 142.3	247.9 ± 163.2	339.3 ± 144.8	346.2 ± 171.6	301.9 ± 230.9
PCo_2_ (mmHg)							
ECMO =	37.6 ± 7.9	29.9 ± 9.9	30.1 ± 5.9	24.7 ± 6.4	27.4 ± 1.8	25.6 ± 7.5	28.1 ± 10.4
Impella =	38.6 ± 9.8	49.0 ± 16.2	49.9 ± 18.6	51.0 ± 23.9	50.3 ± 10.9 *	42.9 ± 13.3	61.1 ± 16.9
Lactate (mmol/L)							
ECMO =	1.3 ± 0.6	7.2 ± 1.7	7.2 ± 1.1	7.9 ± 2.3	9.1 ± 1.9	8.4 ± 1.7	6.9 ± 2.4
Impella =	2.4 ± 0.6	9.9 ± 2.3	9.9 ± 1.8	10.7 ± 2.5	10.9 ± 1.0	9.9 ± 1.3	9.6 ± 0.3
pH							
ECMO =	7.45 ± 0.09	7.29 ± 0.07	7.23 ± 0.14	7.26 ± 0.15	7.29 ± 0.04	7.34 ± 0.14	7.34 ± 0.21
Impella =	7.48 ± 0.09	7.13 ± 0.16	7.09 ± 0.09	7.07 ± 0.12	7.08 ± 0.08	7.15 ± 0.08	7.05 ± 0.10

Mean ± Standard deviation; * *p* = 0.03, for all other comparison *p* > 0.05.

## Data Availability

All the data supporting reported results can be provided upon request to the corresponding author.

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
