# Peer review of "Severe Myocardial Dysfunction after Non-Ischemic Cardiac Arrest: Effectiveness of Percutaneous Assist Devices"

_jcm, 2021, doi:10.3390/jcm10163623_

Round 1

Reviewer 1 Report

Thank you for the opportunity to review this pre-clinical study on percutaneous assist devices for cardiopulmonary resuscitation. Some overall/ major comments:

  • The introduction is rather long and could be shortened significantly.
  • The authors refer to 2000 guidelines and do not use defibrillation every 2 minutes in the study design. This appears to be a significant limitation and deserves some mention.
  • There are several resent clinical studies on the use of ECMO and Impella for cardiac arrest patients (Yannopoulos D, Lancet 2020, DOI: 10.1016/S0140-6736(20)32338-2, Panagides, J Clin Med. 2021, doi: 10.3390/jcm10020339, and Mørk SR, crit care 2021, doi: 10.1186/s13054-021-03606-5). In the light of these clinical studies showing favorable outcomes with assist devices for cardiac arrest patients, it is a bit unclear how this rather small pre-clinical study adds to the literature. This deserves some focus in the introduction and the discussion.

Some specific comments:

  • Abstract: The sentence stating that epinephrine has recently been questioned does not reflect recent evidence showing that epinephrine does improve ROSC and survival to hospital discharge (paramedic 2 trial and Holmberg et al, Resuscitation 2019).
  • Abstract: the sentence that no significant difference was observed between the 2 ventricular assist devices is a bit ambiguous. For what endpoints were no difference observed?
  • Abstract conclusion: In the methods section it is stated that it is an exploratory study. Thus, the study cannot infer on a causal effect but should reflect that an association was seen.
  • Introduction lines 70-77: The notion about epinephrine and the PARAMEDIC2 trial seems unnecessary and also very unbalanced. The PARAMEDIC2 trial was not powered to show an effect on survival to hospital discharge or neurological outcomes. However, the trial showed that use of epinephrine did improve ROSC and survival to hospital discharge. Thus, it was a positive trial leading to ILCOR recommending the use of epinephrine. Repeated epinephrine boluses are always associated with worse outcomes due to resuscitation time bias. Suggest omitting this part to shorten the introduction.

Methods:

  • What breed and sex of pigs were used? Please describe.
  • Were any exclusion criteria applied? Please describe.
  • How was randomization performed? With 17 pigs, one would assume the randomization to come out with 6-6-5 in each group? Please describe.
  • Line 115: What kind of anesthesia was used and how much? Please describe.
  • Lines 142-144: Please describe the rate and depth of the compressions performed using the piston device.
  • Line 147: What kind of defibrillator was used? Moreover, what was the energy level for subsequent shocks? Please describe.
  • Lines 146-148: The authors describe that defibrillation was performed every 3 minutes and atropine was used and describe that resuscitation was performed according to guidelines with references to guidelines from 2000. Thus, it appears as the resuscitation was carried out according to outdated guidelines – why?
  • How was ROSC defined? Please describe.
  • Overall, it is a bit unclear when the intervention was stopped? Did ECMO flow continue after ROSC or was it stopped when ROSC was obtained? Please specify.
  • Lines 160-161: It appears as fluid was used but no inotropy which does not reflect clinical practice in an ICU. Please describe the rationale.
  • Lines 180-181: The sentence that it was conducted as a hypothesis-generating study is a bit vague. What does it mean? Does it mean that no sample size calculation was performed?

Results:

  • It appears as the intervention groups received less shocks when compared to the control group although the VF duration seemed identical. How come this discrepancy?
  • Table 1: It appears that a “S” is missing: compression -> compressions.
  • Table 3: It appears as the impella group had a much lower ph driven by a greater pCO2. Was this significantly different? Please comment.

Discussion:

  • Lines 234-236: Even though ECMO is no miracle, it does show substantially better survival outcomes when compared to standard CPR. Thus, the statement seems a little miscrediting for a procedure that appears to be a substantial improvement to standard CPR/ post-arrest care.
  • There are important and recent clinical studies on ECMO and Impella that deserves mention and discussion (Yannopoulos D, Lancet 2020, DOI: 10.1016/S0140-6736(20)32338-2, Panagides, J Clin Med. 2021, doi: 10.3390/jcm10020339, and Mørk SR, crit care 2021, doi: 10.1186/s13054-021-03606-5). How does this study add to the existing studies and knowledge on Impella and ECMO?
  • Limitations: This is a VF model, and we cannot infer on other causes/ models of cardiac arrest.

Conclusion:

  • As this is an exploratory study, I suggest rephrasing to: In this porcine model of VF cardiac arrest, early assistance with cardiac devices was associated with improved rates of return of spontaneous circulation.
  • The last sentence of more research needed is obsolete.

Reviewer 2 Report

Manzo-Silberman and colleagues present an animal study looking at resuscitation of sudden cardiac arrest. They studied juvenile pigs, and induced VF with extremely rapid pacing. Animals were allowed to fibrillate for 12 minutes without oxygenation or CPR to induce a severe injury.  The animals were randomized to conventional CPR (with a compressor device) versus CPR with one of two percutaneous support approaches--- VA ECMO or Impella CP insertion. The results were that most unsupported animals did not achieve return of spontaneous circulation (ROSC) (1/5 control animals achieved ROSC) whereas all of the Impella supported animals (5/5) achieved ROSC, and 5/7 ECMO treated animals achieved ROSC.

Overall the paper is well done and the experiments interesting, and as stated by the authors hypothesis generating. There are choices during the experiments that limit the utility of the results, including lack of ventilation during the 12 minute asystolic period but then ventilation with 21 % oxygen which disadvantages the Impella group which has no oxygenator. The ECMO group had (as expected) much higher PO2 values. In addition, the lactate was markedly elevated in all animals even with ECMO, and didn’t appreciably change indicating that the animals were still poorly perfused. The experiments are interesting, but it would have been better to treat acid base disturbances, and adjust oxygenation based on blood gases as would be done in a real patient situation.

We can conclude as well that even in an experimental situation, both ECMO and Impella are associated with significant vascular risks with a significant rate of hemorrhage and vascular injury though the environment is controlled and the “patient” is a sedated laboratory large animal.

Suggestions

Abstract—would refer to “implantation of an intravascular left ventricular”--- would use percutaneous rather than intravascular as is used in the manuscript.

Line 82—“The need an”=--- would put “The need of an…”

Materials and Methods—were sheaths placed in the artery and vein femorally prior to VF or where vessels cannulated during CPR? Was ultrasound available and micropuncture technique used?

Experimental protocol—21 % oxygen during resuscitation doesn’t mirror real practice where usually an oxygen cylinder is used with an Ambu-bag and a much higher Fi02 is delivered—would mention in limitation section

Data Analysis--- is 12 cc of contrast sufficient to fill the LV of this size animal? Why was echocardiography not used? “Telediastolic” could be changed to end-diastolic

Limitations--- it is stated all pigs had normal myocardium without cad but one pig had severe hypertrophy per prior statements—please reconcile—did one animal have severe hypertrophy pre-existing and how common is this in pigs?

Round 2

Reviewer 1 Report

Thank you for the revised manuscript on the effectiveness of 2 percutaneous assist devices for cardiopulmonary resuscitation. The manuscript has been significantly improved. However, there are several elements of the methods section that are still not clearly described in the manuscript. Moreover, the manuscript would benefit from some overall description of why it has a primary endpoint that does not answer the aim of the study? The abstract states that the goal was to compare the efficacy of two left ventricular assist devices for treating myocardial stunning, the introduction states that the aim was to compare hemodynamic parameters, myocardial recovery, and peripheral organ damage – but the primary endpoint is ROSC?

Some specific comments:

  • Abstract aim: It is described that the study aims to treat myocardial stunning. However, no measure of myocardial stunning is reported in the conclusion. Aim and conclusion should be more aligned.

  • Line 98: “the Danish study” seems to imply that everyone would be thinking about the same Danish study. I think it would be more correct to state “a Danish multicenter study”.

  • Please describe that no exclusion criteria were applied

  • It is still unclear how the randomization was performed. Did the authors use block randomization, or simple randomization and was it performed using a computer program? Please describe in the manuscript.

  • Lines 142-144: Please describe the rate and depth of the compressions performed in the manuscript. Moreover, please elaborate on whether the device compressed with 4, 5 or 6 cm during CPR and whether all animals received compressions with the same depth?

  • Line 147: It is still not clear what kind of external defibrillator was used? Did the authors use e.g. Zoll, Lifepak, or Schiller? Was it a biphasic waveform? This is relevant as 200J does not have the same converting potential depending on the type of defibrillator. Moreover, what was the energy of subsequent shocks? Please describe in the manuscript.

  • It is still unclear how ROSC was defined? How high should the blood pressure be for it to be considered “with a pulse”, and for how long? Did the authors use the definitions applied for in-hospital cardiac arrest, i.e. 20 minutes? Please describe in the manuscript.

  • Lines 290-292: The authors argue that 12 minutes would reflect the delay of the EMS for OHCA. However, the delay in applying the devices would likely be much longer. This deserves mention in the discussion.

  • Limitations: It should be acknowledged that this is a hypothesis generating study without any sample size calculation. Moreover, it should be acknowledged that the resuscitation does not reflect current guideline recommendations.

Author Response

please see the attachement
